# Vaccination conspiracy beliefs among social science & humanities and STEM educated people—An analysis of the mediation paths

Željko Pavić[1]☯, Adrijana Šuljok[ID][2]☯*

**1** Faculty of Humanities and Social Sciences, Josip Juraj Strossmayer University of Osijek, Osijek, Croatia,
**2** Institute for Social Research in Zagreb, Zagreb, Croatia

☯ These authors contributed equally to this work.
* adrijana@idi.hr

**Data Availability Statement:** All relevant data are within the manuscript and its Supporting information files.

## Abstract

Understanding vaccine hesitancy is becoming increasingly important, especially after the global outbreak of COVID-19. The main goal of this study was to explore the differences in vaccination conspiracy beliefs between people with a university degree coming from different scientific fields—Social Sciences & Humanities (SH) and Science, Technology, Engineering and Mathematics (STEM). The study was conducted on an online convenience sample of respondents with college and university degrees in Croatia (N = 577). The results revealed that respondents educated in SH proved to be more prone to vaccination conspiracy beliefs. The indirect effect through science literacy was confirmed, while this was not the case for the indirect effects through health beliefs (natural immunity beliefs) and trust in the healthcare system. However, all three variables were important direct predictors of vaccination conspiracy beliefs. Female gender and religiosity were positively correlated with vaccination conspiracy beliefs, while age was not a statistically significant predictor. The authors concluded by emphasizing the necessity of the more theoretically elaborated approaches to the study of the educational and other socio-demographic differences in vaccine hesitancy.

## Introduction

It has become commonplace to start a research paper that deals with vaccine hesitancy with a paradox that stems from the tension between the well-known and scientifically-proven benefits of vaccination and the rising tide of vaccine hesitancy. The paradox has led to an extensive corpus of research and various models that aimed to explain and systematize the main factors that lie behind vaccine hesitancy. For instance, according to the so-called 3Cs model proposed by the WHO EURO Vaccine Communications Working Group in 2011 and further elaborated by the SAGE Working Group on Vaccine Hesitancy and MacDonald [1], the factors can be divided into three main groups: (a) confidence (trust in vaccine safety and effectiveness, trust in healthcare providers and policy-makers), (b) complacency (health beliefs, perception of vaccine-preventable diseases as risk-free, thus making vaccination unnecessary), and (c) convenience (availability, affordability, accessibility, science and health literacy, etc.). Research

**Funding:** The author(s) disclosed receipt of the following financial support for the research, authorship and/or publication of this article: This work was funded by Croatian Science Foundation (grant number: HRZZ IP-2019-04-7902). Željko Pavić is principal investigator, Adrijana Šuljok is a member of the research group. The funders had no role in study design, data collection and analysis, decision to publish, or preparation of the manuscript. https://hrzz.hr/en/.

**Competing interests:** The authors have declared that no competing interests exist.

confirmed the importance of the elements from the 3Cs model [e.g. 2, 3]. Given the focus of the current study and the multitude of research studies, we will only outline the main findings of selected elements from the 3Cs model pertaining to the institutional trust (confidence), postmodern health beliefs (complacency), and science literacy (convenience—"ability to understand").

When it comes to the institutional trust, a great number of research studies established a link between trust in social institutions and the healthcare system on one side, and vaccine hesitancy on the other, especially in incident/critical situations. For instance, Lewis and Speers [4] linked the confidence crisis in the MMR vaccine in the UK with the lack of confidence in the executive power at the time, and Mesch and Schwirian [5] found that willingness to vaccinate against H1N1 in the USA during the 2008/2009 pandemic was negatively linked to confidence in the government's ability to address this health crisis. Orr et al. [6] studying the example of the 2013 Polio outbreak in Israel, found that two of the five most important reasons to resist vaccination were related to trust in the government (general mistrust and the conviction that the government had obtained too many vaccines and wanted to get rid of them by vaccinating children). Similar conclusions about the importance of trust were put forward in other studies [7–11].

Complacency also contributes to low vaccine acceptance [2]. With regard to the so-called postmodern health beliefs (preference for natural immunity, hesitancy with regard to the use of pharmaceuticals, etc.), Repalust et al. [12] found that the use of complementary and alternative medicine (CAM) was positively associated with vaccine hesitancy in Croatia. Similar correlation was established in the research of Pavić and Milanović [13], where postmodern health beliefs were associated with CAM use and suspicion in conventional medicine in Croatia, whereas other research also confirms the link between such type of health beliefs and vaccine hesitancy [14–18]. On the other hand, as shown by Lorini et al. [19], the impact of health literacy, as a specific type of science literacy, on vaccine hesitancy is not clear, given that reliable studies which dealt with the topic were not numerous and that the studies used different measures and brought about mutually opposing results that depend on age, country and vaccine types.

Generally speaking, the relationship between educational level and vaccine hesitancy is non-consistent and thus non-conclusive, even though it seems that most of the research results confirmed negative correlation between educational level and vaccine hesitancy. For example, such results were obtained by Ritvo et al. [20] on a sample of the Canadian general population and Peretti-Wattel et al. [21] in a study conducted in France. On the other hand, Börjesson and Enander [22] determined that in Sweden it was the less educated who were more likely to be vaccinated, while Casiday et al. [23] did not establish the correlation between acceptance of MMR vaccine and education. Numerous other studies can be cited that shows a positive [24, 25] or negative [6, 26, 27] correlation between educational level and pro-vaccination attitudes and behaviors. However, such studies explore educational level in general, probably tacitly assuming that science, health or vaccine literacy/knowledge was the mediating mechanism.

In the current study, we wanted to go beyond just exploring the level of education itself and its correlation to vaccine hesitancy. Bearing in mind the possible epistemic differences between the fields of science, we aimed to examine whether differences in the fields of education (i.e., scientific fields) might influence vaccine hesitancy. In other words, our expectation was that the academic background of individuals educated in social sciences and humanities (hereinafter: SH), that are also known as "soft" and cognitively non-restrictive sciences, differed from the academic background of the individuals educated in natural science, technology, engineering and mathematics (hereinafter: STEM) fields, usually defined as "hard", cognitively restrictive scientific fields [28]. Consequently, there should be different levels of vaccine hesitancy

among individuals with a degree in these fields. Namely, our basic contention was that individuals educated in SH fields might be somewhat more skeptical, that is, more inclined to relativize and point out uncertainties relating to vaccines than individuals educated in STEM fields. Such differences can be the result of the different academic socializations and/or the individual characteristics which are important when choosing a field of study. Additionally, it is our contention that such differences will be mediated by some of the indicators coming from the 3Cs model (trust in the healthcare system, specific health beliefs, and science literacy). The basis for this assumption lies in the concept of socio-cognitive differentiation of scientific fields and their cultures, that is, in the epistemic differences between them, which we will elaborate in the next section of the paper. In addition, up to this moment, there are only few studies which dealt with this research topic, mainly confirming that vaccine hesitancy is higher among SH graduates [29–31]. However, the factors behind the established differences are completely unexplored.

Therefore, we proceed with a summary of possible epistemic differences between social sciences and humanities on one side, and science, technology, engineering and mathematics on the other. Then we link these differences to the aforementioned 3Cs factors influencing vaccine hesitancy. In short, we aim to explore selected 3Cs indicators as possible mediating mechanisms arising from the epistemic differences between the scientific fields.

## Epistemic differences and vaccine hesitancy

When it comes to fields of science, there have been many disputes in relation to (1) importance and legitimacy of knowledge in different fields, (2) methodological and epistemic standards and their possible unification, and (3) objectivity of the scientific knowledge in general. Some of these differences between the scientific fields can be grounded in a concept called „epistemic cultures"[32] or related concepts that indicate the socio-cognitive differentiation of sciences, that is, scientific fields [33], Becker's academic tribes and territories [34], Biglan's distinction of soft and hard scientific fields [28, 35], etc. These concepts, mostly conceived in the 1970s and 1980s and developed later, deal with differences in cognitive "objects", "styles", "structures" and "cultures". If we try to make a rough cross-section of these theoretical approaches, it could be argued that STEM fields are characterized by a more rigid cognitive style, routine research and protocolisation, epistemological realism, an emphasis on the objectivity and superiority of scientific methods and practices, and an understanding of scientific knowledge as a cumulative endeavor. On the other hand, SH fields are characterized by theoretical pluralism, discursiveness, less rigid cognitive style, more open and uncertain findings of research, that is, epistemological relativism and greater criticism of scientific methods and practices.

To illustrate the aforementioned differences, a study conducted on a sample of Croatian scientists in natural and social sciences speaks in support of the differences in the understanding of scientific objectivity between the different fields [36]. In this study it was established that natural scientists were firmly convinced of the objectivity of their disciplines, and that they had high confidence in scientific methods and procedures, wherein they emphasized reproducibility and measurement. In contrast, the author of the study noted that relativism was more common among social scientists, who more often pointed out that subjectivity was somewhat inevitable, showed greater skepticism about the omnipotence of research rules and methods, are more often suspected of the notion that objectivity can be achieved, or even in principle deny the possibility of objectivity [36]. In other words, the author summarized that the inclination to positivism and the idea of the fully objective science can be more often found among researchers coming from natural sciences. On the other hand, different ideas and cultural values were more often present in social sciences and humanities. By definition,

these sciences explore the human being and human societies, thus encouraging self-awareness, self-exploration and self-expression that reject the exclusive domination of the medical science (and natural sciences in general) when it comes to delineating what is and what is not a legitimate health issue. On the other hand, STEM educated people are socialized into an epistemic culture that promotes the more objective epistemic standards. For instance, biomedicine-trained individuals embrace the experimental epistemic culture as the „gold standard", thus often rejecting the social sciences methods, such as surveys and qualitative methods, as insufficiently rigorous [37].

The differences in vaccine hesitancy might be partly explained by a different role that trust and criticism play in the STEM and SH epistemic cultures. Social sciences and humanities in general are characterized by epistemic cultures with high levels of context-dependence and with paying close attention to complexity and uncertainty. In consequence, it can be assumed that the people with an SH degree would be more prone to criticism in relation to scientific objectivity, and therefore less willing to accept a position of a layman who accepts the scientific results without a critical stance. When it comes to the issue of trust in social institutions, as Kuhn notes [38], social sciences find themselves in an ambivalent position as related to the social elites. On the one hand, social sciences share some metaphysical assumptions, such as the assumption of man's unsocial nature that must be socialized and tamed in order to produce a good society. On the other hand, social sciences reflect critically on current social processes and acknowledge the discrepancy between the social "facts" and the ideal and promised society, wherein particular social institutions and social elites can be heavily criticized for not delivering social services in the expected way.

## Research goal and hypotheses

The main goal of the current study was to connect the aforementioned differences between the „hard"and „soft"sciences with vaccine hesitancy. To be more precise, the hitherto research of vaccine hesitancy explored the impact of educational level only in general. However, the above-mentioned considerations lead us to believe that the difference in epistemic cultures might exert a significant impact on vaccine hesitancy, in particular vaccination conspiracy beliefs. Therefore, our goal was to fill in the gaps of the research so far by determining possible differences between science fields/fields of study when it comes to vaccine hesitancy, and to explore proximate cause of such differences. With this in mind, our aim was to explore possible differences in vaccination conspiracy beliefs as a specific type of vaccine hesitancy between university educated people graduating in different science fields. Consequently, our hypotheses were stated as follows:

H1. Respondents who graduated in STEM fields have lower vaccination conspiracy beliefs in comparison to the respondents who graduated in SH fields.

H2. Trust in healthcare system, natural immunity beliefs and science literacy are the mediators of the differences in vaccination conspiracy beliefs described in H1.

Our H2 represents a parallel multiple mediation model which can be visualized by means of the following conceptual diagram (Fig 1). Here we emphasize that in order to preserve visual clarity we omitted control variables (gender, age, and religiosity) from the figure. In other words, it is important to note that the mediation from H2 is estimated while simultaneously controlling the above-mentioned demographic variables. Consequently, the possible mediating influences cannot be attributed to the confounding effects of the demographic variables since they are controlled for in the statistical model.

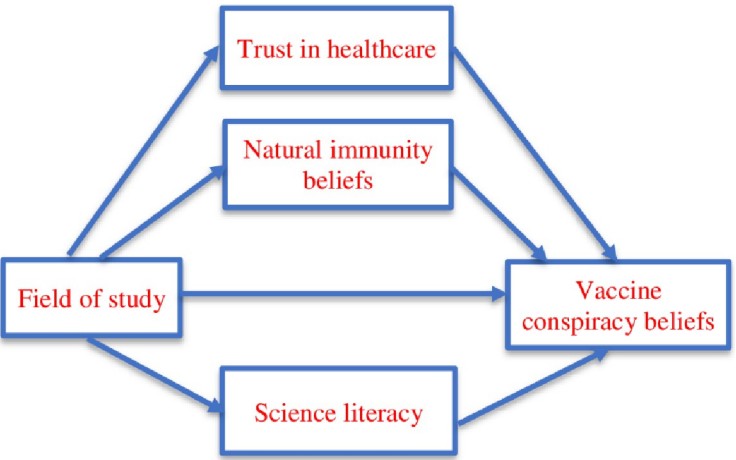

**Fig 1. Conceptual diagram of the parallel mediation tested in the study.**

## Methods and instruments

A convenience sample consisted of college and university educated people in Croatia (N = 577), and an online questionnaire was used as the data-collection tool. The participants were recruited with snowball method. The initial sample size was 624, but 47 respondents (7.53%) had missing values on some of the study variables. We conduced binary logistic regression with the missing value as the criterion variable and did not find any significant relationship with other study variables. That is why we proceeded with the complete case analysis, which reduced our sample from 624 to 577 respondents. The data were collected in June 2019 within a pilot study of vaccine hesitancy conducted by the authors. Namely, only the answers of the respondents from the pilot study who graduated from a university were used in the current study. The sample comprised 345 women and 232 men (59.79% and 40.21%, respectively). There were 315 SH and 262 STEM graduates (54.59% and 45.41%).

As for the delineation of SH and STEM fields, the official classification by the Ministry of Science and Education of the Republic of Croatia was used [39].

Informed consent was obtained by means of a written online form. Participants could indicate that they have read the description of the study, were over the age of 18, and that they agreed to participate. Participants did not receive any compensation for completing the survey, Members of the Ethics Committee of the Institute for Social Research Zagreb agreed that the project proposal (Protocol 1112/2017) was in accordance with ethical principles and rules of conduct highlighted in the ISRZ Code of Ethics, as well as the rules of conduct of research in social sciences in general.

As for the measurement scales, vaccination conspiracy beliefs were measured by the scale constructed by Shapiro et al. [40], which comprises seven items. All items were answered on a 5-point scale ranging from 1 = completely disagree to 5 = completely agree. The descriptive statistics for the scale items are presented in Table 1. The results from the items were added in order to get the total summary score (M = 15.11, SD = 8.44; $M_{SH}$ = 16.27; $SD_{SH}$ = 8.83; $M_{STEM}$ = 13.72; $SD_{STEM}$ = 7.74; t = 3.70: p = 0.00) to be used in the subsequent analyses with higher values representing higher belief in vaccination conspiracies. Internal consistency of the scale as the measure of reliability (Cronbach's alpha) was 0.967.

**Table 1. Descriptive statistics for the vaccine conspiracy beliefs scale.**

| Item | M (SD) | Corrected Item-Total Correlation | Skewness | Kurtosis |
|---|---|---|---|---|
| Vaccine safety data is often fabricated | 2.38 (1.32) | .78 | .70 | -.66 |
| Immunizing children is harmful and this fact is covered up. | 1.92 (1.28) | .91 | 1.29 | .42 |
| Pharmaceutical companies cover up the dangers of vaccines. | 2.53 (1.37) | .87 | .55 | -.95 |
| People are deceived about vaccine efficacy. | 2.04 (1.30) | .91 | 1.08 | -.08 |
| Vaccine efficacy data is often fabricated. | 2.05 (1.29) | .89 | 1.09 | .00 |
| People are deceived about vaccine safety. | 2.24 (1.34) | .93 | .80 | -.65 |
| The government is trying to cover up the link between vaccines and autism. | 1.96 (1.31) | .89 | .10 | .08 |

A three-item scale of natural immunity beliefs was taken from the Martin and Petrie [41], Vaccination Attitudes Examination (VAX) scale. Those items were answered on a five-point scale ranging from 1 = completely disagree to 5 = completely agree, as well. These three items were also summed in order to get the total summary score (M = 6.60, SD = 3.63; $M_{SH}$ = 6.92; $SD_{SH}$ = 3.83; $M_{STEM}$ = 6.23; $SD_{STEM}$ = 3.34; t = 2.32; p = 0.02) since the internal consistency, as measured by Cronbach's alpha, was high (0.93). Higher values represent higher belief in natural immunity. The descriptive statistics for the items are shown in Table 2.

The trust in healthcare system was measured by using a revised version of the nine-item scale constructed by Shea et al. [42], To be more specific, we left out the item about the equal treatment of patients of all races and ethnicities given that the Croatian society is largely mono-ethnic and mono-racial. All the items were answered on a five-point scale ranging from 1 = completely disagree to 5 = completely agree. Even though Shea et al. [42] confirmed that the scale can be divided into two sub-scales (value congruence and competence), in the current study this scale proved to be one-dimensional with a high level of reliability (Cronbach's alpha was equal to 0.90). The descriptive statistics for the items is shown in Table 3, whereas the total summary score was also used in subsequent analyses (M = 23.82, SD = 6.37; $M_{SH}$ = 23.28; $SD_{SH}$ = 6.34; $M_{STEM}$ = 24.46; $SD_{STEM}$ = 6.35; t = -2.22; p = 0.03) with higher values representing higher trust in healthcare system.

In the current study the items from the so-called Oxford scale of science literacy [43, 44] were used. The motivation for using a general science literacy scale instead of constructing a specific vaccine knowledge scale lies in our judgement that the specific scale would have to be composed of items that are themselves too „contaminated"with attitudes towards vaccines. For instance, the scale developed and applied by Cvjetkovic et al. [29] contained items about the connection between vaccines and autism and about the dangers of applying multiple vaccines at the same time. These items, although relevant, are strongly intertwined with vaccine attitudes and can be thought of more as a measure of vaccine attitudes than of knowledge. In

**Table 2. Descriptive statistics for the natural immunity beliefs scale.**

| Item | M (SD) | Corrected Item-Total Correlation | Skewness | Kurtosis |
|---|---|---|---|---|
| Natural immunity lasts longer than a vaccination | 2.32 (1.37) | .83 | .70 | -.66 |
| Natural exposure to viruses and germs gives the safest protection | 2.26 (1.28) | .87 | 1.29 | .42 |
| Being exposed to diseases naturally is safer for the immune system than being exposed through vaccination | 2.03 (1.23) | .87 | .55 | -.95 |

**Table 3. Descriptive statistics for the trust in healthcare system scale.**

| Item | M (SD) | Corrected Item-Total Correlation | Skewness | Kurtosis |
|---|---|---|---|---|
| The Health Care System does its best to make patients' health better | 2.74 (1.13) | .67 | .17 | -.86 |
| The Health Care System covers up its mistakes* | 3.57 (1.00) | .65 | -.56 | -.07 |
| Patients receive high quality medical care from the Health Care System | 2.98 (0.97) | .68 | -.31 | -.59 |
| The Health Care System makes too many mistakes* | 3.15 (0.97) | .70 | .04 | -.38 |
| The Health Care System puts making money above patients' needs* | 3.15 (1.09) | .70 | -.01 | -.73 |
| The Health Care System gives excellent medical care | 2.58 (0.97) | .68 | .02 | -.68 |
| The Health Care System lies to make money* | 2.52 (1.09) | .73 | .52 | -.23 |
| The Health Care System experiments on patients without them knowing* | 2.10 (1.17) | .60 | .84 | -.22 |

* Reversely scored items.

our view, the Oxford scale measures „non-problematic"items and therefore, in spite of its general nature, is a more valid measure of science knowledge in our case. The Oxford scale arose as a joint effort of several researchers [43, 44] used in the report *Science and Engineering Indicators* prepared by the U.S. National Science Board. Some of the questions were also used in several Eurobarometer surveys [45, 46]. The scale consisted of 15 questions that probe into knowledge about general science facts, and the respondents have to mark them as „true", „untrue", „don't know"answer. Both incorrect and „don't know"answers were merged into one category, while correct answers comprised the other category making this variable dichotomous. The frequencies of answers on individual items are given in Table 4. The results were summed in order to obtain the total score (M = 12.30, SD = 2.42; $M_{SH}$ = 11.65; $SD_{SH}$ = 2.60; $M_{STEM}$ = 13.00; $SD_{STEM}$ = 2.02; t = -6.82; p = 0.00), with higher values indicating higher science literacy. Internal consistency of the variable (Kuder-Richardson—KR20—coefficient) was reasonably high (0.70).

**Table 4. Descriptive statistics for the science literacy scale.**

| Item | Correct answer (%) | Incorrect answer, don't know answer (%) |
|---|---|---|
| Antibiotics kill viruses as well as bacteria | 87.69 | 12.31 |
| The Sun goes around the Earth | 83.54 | 16.46 |
| The center of the Earth is very hot | 90.81 | 9.19 |
| The oxygen we breathe comes from plants | 88.21 | 11.79 |
| The earliest human beings lived at the same time as the dinosaurs | 85.44 | 14.56 |
| By consuming genetically modified fruit we alter our genes | 72.77 | 27.73 |
| All radioactivity is man-made | 85.96 | 14.04 |
| It is the mother's genes that decide whether the baby is a boy or a girl | 75.91 | 24.09 |
| More than half of human genes are identical to those of mice | 50.26 | 49.74 |
| Electrons are smaller than atoms | 85.44 | 14.56 |
| Lasers work by focusing sound waves | 66.38 | 33.62 |
| It takes one month for the Earth to go around the Sun | 90.47 | 9.53 |
| Radioactive milk can be made safe by boiling it | 88.39 | 11.61 |
| The continents on which we live have been moving their location for millions of years and will continue to move in the future | 93.93 | 6.07 |
| Human beings, as we know them today, developed from earlier species of animals | 81.63 | 18.37 |

**Table 5. Intercorrelation matrix (Pearson's correlation).**

| Variable | Gender | Age | Religious identification | Field of study | Natural immunity beliefs | Trust in healthcare | Scientific literacy | Vaccine conspiracy |
|---|---|---|---|---|---|---|---|---|
| Gender | 1 | -.08* | . 21** | - .17** | .17** | - .12** | - .26** | .18** |
| Age | - .08* | 1 | .15** | - .03 | - .07 | . 05 | .06 | - .04 |
| Religious identification | . 21** | .15** | 1 | - .02 | .31** | - .21** | - .32** | .36** |
| Field of study | - .17** | - .03 | - .02 | 1 | - .10** | .09* | .27** | - .15** |
| Natural immunity beliefs | .17** | - .07 | .31** | - .10** | 1 | - .55** | - .45** | . 81** |
| Trust in healthcare | - .12** | . 05 | - .21** | .09* | - .55** | 1 | .31** | - .66** |
| Scientific literacy | - .26** | .06 | - .32** | .27** | - .45** | .31** | 1 | - .48** |
| Vaccine conspiracy | .18** | - .04 | .36** | - .15** | . 81** | - .66** | - .48** | 1 |

Notes:

*p < .05,

**p < .01;

Gender: 0—Male, 1—Female; Field of study: 1—SH, 2—STEM.

Finally, sociodemographic variables like age, gender, religiosity were also collected. Age was measured by asking participants' year of birth and then re-coding it into the age measured in years. Gender was measured as female and male, while religious identification was measured from 1-non religious to 6- very religious. Average age was 39.01 years (min = 20; max = 73; SD = 9.25), while the average result on the 1–6 religiosity scale was 3.03 (SD = 1.61).

All the instruments were translated into Croatian, and then back-translated in order to check the validity of the translation. Appropriate changes were made accordingly.

## Results

With the purpose of gaining an initial insight into the bivariate relations between the study variables, in Table 5 we present the intercorrelation matrix. Among other things, it can be observed that the higher level of vaccine conspiracy beliefs can be found among women (r = 0.18; females = 16.33, males = 13.29, Cohen's d = 0.37), more religious people (r = 0.36), people with higher beliefs in natural immunity (r = 0.81), people with lower trust in healthcare system (r = —0.66), and people with lower scientific literacy (r = —0.48). With respect to the field of study (H1), participants with degrees in social sciences and humanities are more likely to have vaccine conspiracy beliefs (r = 0.18; SH = 16.27, STEM = 13.72, Cohen's d = 0.31).

In order to test total, direct and indirect (mediation) effects of the field of study on vaccine conspiracy beliefs (H2), we used PROCESS, a mediation and moderation SPSS macro written by Hayes [47]. Given that PROCESS is a regression-based model, first we present the regression analysis with conspiracy beliefs as the criterion variable (Table 6).

In comparison to SH graduates STEM graduates score 0.70 points lower on the vaccine conspiracy beliefs scale. A one-unit increase in the natural immunity beliefs on average raises the results on the vaccine conspiracy belief scale by 1.34 points when all other variables are held constant. A one-unit increase in trust in the healthcare system on average lowers the results on the vaccine conspiracy belief scale by 0.40 points. Finally, a one unit increase in science literacy decreases vaccine conspiracy belief by 0.33 points.

As for the demographic variables (gender, age, and religious identification), we can observe that religious identification is a statistically significant predictor, whereas age and gender are

**Table 6. Linear regression analysis with the result on the vaccine conspiracy beliefs scale as the criterion variable.**

| Variable | B | SE | t | p | LLCI | ULCI |
|---|---|---|---|---|---|---|
| Intercept | 17.90 | 1.90 | 9.41 | .00 | 14.16 | 21.64 |
| Gender | - .10 | .39 | - .26 | .79 | - .87 | .66 |
| Age (in years) | .03 | .02 | 1.27 | .20 | - .02 | .06 |
| Religious identification | .50 | .12 | 4.05 | .00 | .26 | .74 |
| Field of study | - .70 | - .38 | - 1.83 | .07 | - 1.45 | .05 |
| Natural immunity beliefs | 1.34 | .07 | 20.71 | .00 | 1.21 | 1.47 |
| Trust in healthcare system | - .40 | .03 | - 11.07 | .00 | - .45 | - .31 |
| Scientific literacy | - .33 | .09 | - 3.69 | .00 | - .50 | - .15 |

Notes: $R^2 = 0.74$; F = 202.42; p = 0.00; Gender: 0—Male, 1—Female; Field of study: 1—SH, 2—STEM; LLCI—lower limit of the confidence intervals; ULCI—upper limit of the confidence intervals.

not significant predictors when all other variables are entered into equation. Here we can once more emphasize that there is a bivariate correlation between gender and vaccine conspiracy beliefs, with women scoring 3.04 points higher than men (Cohen's d = 0.37), but this difference disappears in the multivariate analysis. We can also note that a one-unit increase in religious identification raises the result on the vaccine conspiracy belief scale by 0.50 points when the other variables are held constant.

After the full regression model, we proceeded with the mediation analysis. In Table 7, total, direct and indirect effects of field of study are presented, with age, gender and religiosity entered as covariates. The covariates were used in order to check for the possible spurious effects of our predictors. As suggested by Hayes [47], bootstrapping was used as a method of obtaining confidence intervals of the estimates of the indirect effects and the pairwise differences between indirect effects.

We can observe that the total effect coefficient amounts to—2.18, meaning that STEM graduates score 2.18 point lower on the vaccine conspiracy beliefs scale when compared to SH graduates, and when both direct and indirect effects are accounted for. The coefficient for total

**Table 7. Vaccine conspiracy beliefs—Mediation analysis.**

| Total effect | | | | |
|---|---|---|---|---|
| Coeff. | SE | t | LLCI | ULCI |
| -2.18 | .66 | - 3.30** | - 3.47 | - 0.88 |
| Direct effect | | | | |
| Coeff. | SE | t | LLCI | ULCI |
| - .70 | .38 | - 1.84 | - 1.45 | .05 |
| Indirect effects | | | | |
| | Coeff | BootSE | BootLLCI | BootULCI |
| Natural immunity beliefs | - .71 | .38 | - 1.48 | .01 |
| Trust in the healthcare system | - .37 | .21 | - .78 | .02 |
| Science literacy | - .39 | .12 | - .63 | - .18 |

Note:

*p < .05,

**p < .01;

LLCI—lower limit of the confidence intervals; ULCI—upper limit of the confidence intervals.

effect can be decomposed into the direct effect (- 0.70) and the indirect (mediation) effect (-1.45), with the latter including indirect effects of natural immunity beliefs (- 0.71), trust in healthcare system (- 0.37) and science literacy (- 0.39). The direct effect is not statistically significant, while only the indirect effect of scientific literacy proved to be significant by means of the bootstrapping analysis. In the remaining two cases, the bootstrapping analysis of indirect (mediating) effects revealed that zero effect is included into the confidence intervals. All bootstrapping confidence intervals of the pairwise differences between mediation effects also contained zero difference, thus not being able to confirm the existence of the pairwise differences between the indirect effects.

## Discussion

In pursuance of the research goal, in this paper we proposed two hypotheses. The first one (H1) was related to the differences in vaccination conspiracy beliefs between SH and STEM fields, and the second one (H2) to the possible mediating mechanism of such differences.

To summarize the findings pertaining to H1, even though both groups mostly largely reject vaccination conspiracy beliefs, our findings confirm that the people with an SH degree are more prone to vaccination conspiracy beliefs than the people with a STEM degree, thus confirming H1. The bivariate difference equals 2.55 points, it falls to 0.70 when demographic variables (gender, age, and religiosity) are accounted for, but it also rises to 2.18 when both direct and indirect effects are included. Bearing in mind the lack of similar studies, we cannot directly compare our results with other research. We can confirm that our results are in concordance with the results obtained by Cvjetkovic et al. [29], who determined that the law students of the Belgrade University (Serbia) had substantially more negative vaccine attitudes than the engineering, and especially medical students, even while controlling for vaccine knowledge. However, since Cvjetkovic et al. did not conduct a full mediation analysis, their results cannot be directly compared to our results with regard to the indirect effects. Likewise, a study conducted by Šálek et al. [31] on a sample of medical and teacher education students in the Czech Republic found differences in the positive vaccine perception rate between those two groups. Vaccine acceptance rate was lower among teacher education students (72%) than among medical students (92%). Also teacher education students more often reported negative experiences with vaccination and were more often alternative medicine followers.

On the other hand, the mediation analysis conducted within our study confirmed the mediation effect of scientific literacy, while the mediation effect of natural immunity beliefs and the trust in healthcare system were not reliably confirmed, thus only partially confirming our second hypothesis (H2). Notwithstanding that fact, we emphasize that all mediation factors are important direct predictors of vaccine conspiracy beliefs. The data employed in the study just did not confirm the mediation function between the field of study and vaccination conspiracy beliefs in case of the two mediators. Another word of caution is also warranted here. As mentioned before, a parallel multiple mediation model with control variables was tested in our study. This means that mediation paths were tested with simultaneously controlling the effects of gender, religiosity and age. In additional analyses we confirmed that all mediation paths were statistically significant when gender was not employed as a control variable, while in that case indirect effects of natural immunity beliefs, trust in the healthcare system and science literacy amount to—0.87,—0.43 and—0.42, respectively. Consequently, the mediation paths can be explained by the disproportionate presence of women in STEM fields and the correlations between gender on one side, and vaccine conspiracy beliefs and the mediating factors on the other side, as visible from the intercorrelation matrix. In other words, all three mediators are indeed significant, but in cases of two of them (trust in healthcare system and natural

immunity beliefs) this is the case only because women on average happen to have lower trust and higher beliefs in natural immunity, as well as higher vaccination conspiracy beliefs, while at the same time being disproportionally more present in social sciences and humanities.

The direct impact of the natural immunity preference on vaccine conspiracy beliefs is consistent with the results of the study of vaccine hesitancy in Croatia [12], which showed that the use of complementary and alternative medicine (CAM) was positively connected to vaccine hesitancy and vaccine refusal, as well as with a research conducted on a sample of Croatian population [13] that showed a negative connection between „postmodern"views on health (natural immunity included) and trust in conventional medicine as opposed to the trust in CAM. Positive attitudes towards CAM and its use partially follow from the general question of the legitimacy of the medical science authority. Namely, as Sointu [48] summarized, CAM users defy the biomedical definition of health by arguing for holistic health that empowers them by providing a sense of agency, control and meaning. Therefore, CAM produces a new selfhood and subjectivity that is directed towards exploring the inner-self. However, in the current study we could not reliably confirm that such differences represent a mediating factor of the established connection between the field of study and vaccination conspiracy beliefs. Namely, although the mediation effect was present in the hypothesized direction, its statistical significance could not be demonstrated by the bootstrapping analysis. Nevertheless, given the theoretical plausibility of such differences in epistemic culture and the fact that the bootstrapping interval included zero only by a small margin, we think that new studies which will employ random sampling are warranted.

Similarly, we also could not reliably confirm the mediation effect of the trust in the healthcare system with our study data. The possible explanation might be that the aforementioned criticism penetrated all fields of study and that it does not explain the differences in vaccine hesitancy. Perhaps we should not overstate the differences/gap between natural sciences and social sciences/humanities, since objectivity is nevertheless an important scientific ideal in (all) sciences [49]. Especially in the social sciences, methods and precision of natural sciences continue to be an inspiration to be emulated for some of the scientists from this field, but who are still aware of a distinctiveness of the social sciences. Conversely, in their qualitative study of science and non-science (i.e. humanities) tertiary educated people, Quinn et al. [50] established that even people who accepted the scientific method of evidence-based medicine tended to exhibit various „habits of mind", such as the mistrust of authority, open-mindedness, skepticism, and rationality/belief systems that led them to engage in the acceptance and use of CAM. Thus, as Quin et al. concluded, scientific literacy can go hand in hand with belief systems that are prone to potential distrust towards science and scientists. In other words, a high level of science knowledge is not always translated into support for science [51], that is, individuals can use knowledge in different ways, according to their pre-existing interests and motivations [52], or educational epistemic culture. Additionally, it should be emphasized that within the scientific fields there are considerable differences among the various disciplines, and that there are various paradigms and methodological approaches within the same discipline. However, despite the fact that our mediation analysis did not confirm the impact of this factor, we think that the idea should be tested in future studies conducted on different samples, given that the bootstrapping interval from our study in this case as well comprised zero only by a small margin.

We posited that the aforementioned epistemic differences and the obvious assumption that people with a degree in SH should be less familiar with the knowledge coming from natural sciences, biomedical sciences included, could also explain the differences in vaccine hesitancy. In their meta-analysis of the connection between science literacy and science attitudes, after controlling for several potential confounding variables, Allum et al. [53] found a „small but

consistent"correlation that gave some support to the deficit model of the relationship between science knowledge and attitudes. A study in Croatia on a representative sample also indicated a significant but weak relationship between the level of scientific literacy and (positive) attitudes toward science [54]. This is consistent with the results of the current study, in that science literacy is a statistically significant predictor of vaccine conspiracy beliefs, as well as the mediator of the SH-STEM differences in vaccine hesitancy. In other words, the results of the current study confirm the above mentioned "small, but consistent" relationship, given a partial mediation of the differences in the vaccine conspiracy beliefs between the STEM and SH fields through science literacy.

Aside from the discussion pertaining to H1 and H2, we shall also present a short discussion of the results with regard to other sociodemographic variables. We should, however, bear in mind that our study was conducted on a sample of higher-educated individuals, thus not being fully comparable to the results obtained in studies on general population research samples.

To start with, we have not confirmed the relationship between age and vaccine conspiracy beliefs. However, this relationship is highly varied and far from being conclusive. For instance, our results differ from the results obtained by Repalust et al. [12] since in their study the younger age groups were less vaccine acceptant in comparison to the older age groups. However, other research can be cited that determined a positive [55], negative [56], or non-existent [57] relation between age and various measures of vaccine hesitancy. These differences might be the result of age differences in relation to education, health beliefs, trust in social institutions etc. However, to our best knowledge, age differences in vaccination attitudes so far have not been comprehensively theoretically explained.

With regard to the established positive connection between religiosity and vaccine conspiracy beliefs, the results of the current study are mainly consistent with the results obtained by other studies [12]. Possible reasons for the relationship between religiosity and negative vaccine attitudes might include a latent conflict between science and religiosity within the Croatian citizens' value system or, more specifically, the framing of biomedical questions as religious issues, dominated by the idea that medical science should not meddle too much into „God's work". More specifically, religious people might object to the fact that human and animal tissues, such as cell lines from aborted fetuses and hydrolyzed gelatin obtained from animals, are being used in the production of some vaccines [58]. As for the Catholic Church, being the dominant religious community in the Croatian society, according to the document drafted in 2005 by the Pontifical Academy for Life, the use of vaccines with the aborted fetuses' tissue constitutes „at least a mediate remote passive material cooperation to the abortion". Catholics are thus obliged to push for morally acceptable alternatives in the vaccine production, even though the use of vaccines should be tolerated in order to save children's lives. Such a situation presents Catholic parents with a „moral coercion"that must be resolved as soon as possible [59].

When it comes to gender, our results are in line with the majority of studies indicating that vaccine hesitancy is more likely to be found among women [17, 60], even though inverse findings can also be found [61]. In addition, the results of our study suggest an interesting finding that the gender composition of scientific fields can be a part of the explanation of their epistemic differences and its impact on vaccine hesitancy. These results are in line with preponderance of women in the CAM, both as users and practitioners [62], as well as with the positive relationship between CAM beliefs and practices and vaccine hesitancy [63].

## Conclusion and limitations

The results of our preliminary study suggest that future research on vaccine hesitancy should differentiate not only between different education levels, but between education/science fields

as well. Namely, the people with a degree in SH proved to be more prone to vaccination conspiracy beliefs, whereas this difference is probably mediated at least by the differences in science literacy. The reasons could probably be found in the different academic socialization and the background characteristics which are important when selecting a field of study. Future studies should also further explore the possible mediators of this difference and establish some more in-depth theoretical explanations for their link with epistemic differences between scientific fields. In other words, when it comes to education and other sociodemographic factors as well, more comprehensive research is warranted that will go beyond simple descriptions of sociodemographic differences in vaccine hesitancy, in order to draw more complete conclusions, as well as possible recommendations in terms of science/health education.

Even though we did not confirm that beliefs in natural immunity and trust in the healthcare system are mediating paths of the vaccination conspiracy beliefs differences between SH and STEM, these constructs proved to be strong predictors of vaccination conspiracy beliefs. In the context of the COVID-19 pandemics this finding suggests that vaccine hesitancy cannot be contested only by providing factual, scientific information on COVID-19 vaccines. Successful COVID-19 crisis management needs to include a trust-building communication, and to address lay beliefs about vaccine and health in general.

In our view, the main weakness of the current study represents the convenience sample employed, as well as the data-collection method (online survey) employed in the study. It can be reasonably assumed that some amount of self-selection among potential study participants might be present in the study. Therefore, we cannot exclude the possibility that some sort of bias is shared between the study variables that might influence their relationship. It might also be assumed that the variance of the measurements was somewhat restricted, that is, the less (or more) vaccine hesitant people might have been more inclined to participate in the survey which might have influenced our results and conclusions. As for the measurements employed in the study, more specific measures of vaccine knowledge might bring different results than general measures of science knowledge such as the Oxford scale used in the current study. In addition, given that our study was cross-sectional and the possible omitted variables bias might have existed, causal mechanisms cannot be fully supported by the data from our study.

## Supporting information

**S1 Data.**
(SAV)

**S1 Highlights.**
(DOCX)

## Author Contributions

**Conceptualization:** Željko Pavić, Adrijana Šuljok.

**Data curation:** Željko Pavić, Adrijana Šuljok.

**Formal analysis:** Željko Pavić, Adrijana Šuljok.

**Funding acquisition:** Željko Pavić.

**Investigation:** Željko Pavić, Adrijana Šuljok.

**Methodology:** Željko Pavić, Adrijana Šuljok.

**Project administration:** Željko Pavić.

**Supervision:** Željko Pavić, Adrijana Šuljok.

**Validation:** Željko Pavić, Adrijana Šuljok.

**Writing – original draft:** Željko Pavić, Adrijana Šuljok.

**Writing – review & editing:** Željko Pavić, Adrijana Šuljok.

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
