## [Decision Letter · Decision Letter 0]

20 Oct 2021

PONE-D-21-19906Vaccination conspiracy beliefs: differences between persons educated in different scientific fields and their mediating mechanismsPLOS ONE

Dear Dr. Šuljok,

Thank you for submitting your manuscript to PLOS ONE. After careful consideration, we feel that it has merit but does not fully meet PLOS ONE’s publication criteria as it currently stands. Therefore, we invite you to submit a revised version of the manuscript that addresses the points raised during the review process.

We look forward to receiving your revised manuscript.

Kind regards,

Kristina Hood, Ph.D.

Academic Editor

PLOS ONE

Journal Requirements:

2. Please amend your manuscript to include your abstract after the title page.

Reviewers' comments:

Reviewer's Responses to Questions

**Comments to the Author**

1. Is the manuscript technically sound, and do the data support the conclusions?

Reviewer #1: Yes

Reviewer #2: Partly

Reviewer #3: Partly

2. Has the statistical analysis been performed appropriately and rigorously? 

Reviewer #1: I Don't Know

Reviewer #2: Yes

Reviewer #3: No

3. Have the authors made all data underlying the findings in their manuscript fully available?

Reviewer #1: Yes

Reviewer #2: Yes

Reviewer #3: No

4. Is the manuscript presented in an intelligible fashion and written in standard English?

Reviewer #1: Yes

Reviewer #2: Yes

Reviewer #3: Yes

5. Review Comments to the Author

Reviewer #1: Thank you for the opportunity to review the manuscript entitled ‘Vaccination conspiracy beliefs: differences between persons educated in different scientific fields and their mediating mechanisms'.

The author(s) present a study in which they investigated vaccine conspiracy beliefs in regard to educational background – social science and humanities versus stem. It is an interesting take on the role of education in vaccine conspiracy beliefs, because education is in this kind of research mostly investigated in terms of educational levels and not field differences.

I found the manuscript to be well written, and it contributes with new insights. Nevertheless, I recommend some changes which can contribute to an even greater quality of it.

Title

I suggest renaming the title to emphasize your main finding and make it more interesting for potential readers, some examples are: ‘Vaccine conspiracy beliefs stronger in social sciences and humanities then in STEM – a correlational study’.

Abstract

Start with an introduction sentence.

Introduction

Page 4, lines 76-79, add references for describing the SH and STEM fields.

Throughout the manuscript replace ‘persons’, perhaps with ‘people’, although I am not sure this is the adequate term too.

Research goals, questions, and methods

Please insert a paragraph that in details describes the procedure of how and when the study was conducted - where was the survey shared, which months and year, how the instruments were translated to Croatian.

Check the journal guidelines – but I find the Descriptive tables should be better put in the Results section. In the Descriptive tables, you can write the SD on items in brackets and italic after the M (e.g., M(Sd) 2.32(1.37))

Replace ‘Reverse items’ with ‘Reversely scored items’ throughout the manuscript.

Results

Add the age range, apart from the M and Sd, as in minimum and maximum.

In Table 6, the significance of the standardized Beta regression coefficients should be written. Now only the statistical significance of B is written.

I would be much more interested to see the differences in the used instruments between the two educational groups then the skewness and kurtosis. So please add another table which will show the M and Sd of the used instruments separately for the SH and STEM subsamples. Even conducting an ANOVA would be appropriate.

Discussion

Start with a sentence that reminds the reader of the aim of the study.

At the end, I also think it is important to state the overall limitations of an online survey, your sample and methodology in general.

Highlights

I suggest rewriting the highlights as more comprehendible or fuller sentences, it is hard for me as a reader to follow it in this form.

Overall

Language and grammar require some editing (especially in the highlights section), so I suggest carefully reading through the manuscript and double checking or engaging a native speaker. I would also emphasize that the variables used here are not the only ones that are shown to be associated with vaccine conspiracy beliefs (another example from Croatia is DOI: 10.1080/08870446.2019.1673894)

Reviewer #2: This article is exploring the differences in vaccination conspiracy beliefs between people belonging to different scientific fields – Social Sciences & Humanities (SH) and Science, Technology, Engineering and Mathematics (STEM). The results revealed that those educated in SH were more prone to vaccination conspiracy beliefs, and this was mediated by health beliefs; lower trust in the healthcare system, and lower science literacy.

Although the authors deal with important questions for the society today, the rationale for this specific research question is not explored enough. The theoretical background need to be more explained, followed by the importance of differentiation between SH and STEM persons. Is there any specific recommendation coming from these findings (in terms of science education)?

Some issues remained unanswered: What kind of professions exactly, do the authors have in the groups under the umbrella of STEM and SH? What are the employment statuses of the respondents? Sometimes people work in completely different fields then those from which they graduated. It would be good to discuss the relation between employment-education status of the respondents and relation of those variables to the conspiracy belief.

Structure of the manuscript in some parts is not adequate for the academic paper. There is no difference between Introduction and Method. Research goals and questions should be placed in the Introduction part. Method part is missing; it should stand alone not incorporated in other parts. Instruments should be placed in the Method part. Sample information is not sufficient: we need to know when the survey was conducted, whether there was ethical approval from any institution, the length of the questionnaire. Also I would recommend reporting the sample characteristics in more details.

Additionally, descriptive statistics are usually part of results not method part. Endorsement of scientific knowledge and vaccine conspiracy belief should be mentioned as research question (because they are presented in the tables), and the total scores could be given separately for SH and STEM part

Tables are not according to APA standards.

I would also like to see empirical confirmation for the assumptions mentioned in the discussion: ...Additionally, a large portion of these differences can be explained by the mediating factors that are part of the 3C's model which, according to our claim, might be stemming from the differences in epistemic cultures, i.e. teaching patterns, cognitive approaches, research traditions, goals, behaviors, logic and methods. How can authors assume that? In the discussion part it would be better if the authors analyze results that they measured in the study.

Finally, what is the relation between Conspiracy beliefs vs. Vaccine intentions? What are the effects of conspiracy belief in the current pandemic? And how can the results of the study be connected to the situation in pandemic – covid-19 conspiracy theories. The connection between vaccine CT and Covid-19 CT should be explained.

Reviewer #3: The goal of the present study was to determine whether were differences in vaccine hesitancy and likelihood of accepting vaccine conspiracy theories in individuals educated in social sciences and humanities versus those educated in STEM. The research question is interesting and timely given concerns related to COVID-19 vaccine hesitancy. However, the rationale is underdeveloped, the method/results lack important details, and the discussion over-reaches. Below, I outline my concerns in more detail.

The introduction is underdeveloped. The authors mention the 3Cs at the beginning of the introduction and review some research on the “confidence” component, but no attention is given to convenience or complacency. They then state that they will examine the 3Cs in relation to those educated in STEM and social sciences/humanities. It is unclear how or why the 3Cs are being examined. More discussion and connection to the research questions would be helpful. The authors also outline differences between STEM and social sciences/humanities fields, but don’t clearly connect this to vaccine hesitancy. It would be helpful for the authors to more explicitly state their reasoning and provide hypotheses, if they had a priori hypotheses. Clearer rationale for the mediation analyses needs to be provided.

More information needs to be provided regarding participant recruitment. How were participants contacted? What was the response rate? Is the sample representative of the population in question? What range of degrees are represented in the sample (e.g., BS/BA, MS/MA, PhD, etc.)? Did participants receive compensation? Was there informed consent?

Were there any missing data, outliers, non-normality, etc.? If so, how were these issues addressed?

Why were age, gender, and religiosity included as covariates? Did results differ if covariates were excluded?

Before reporting the mediation analysis, it would be useful to report simple comparisons (t-tests) indicating whether STEM vs. SH participants differed in any of the mediators or outcome variables.

A clearer description of the mediation analysis(es) is necessary, noting how variables were coded (i.e., STEM vs. SH) and included in the model. I assume a parallel mediation model was estimated.

Remove reference to marginal significance (p = .06). If relying on p-values, an effect is significant or not. Also, discussion of full or partial mediation is inappropriate, given the correlational and cross-sectional nature of the study.

It is unclear to me what purpose the hierarchical regression analysis serves.

The claim in the discussion that the 3Cs mediated difference in education field and vaccine conspiracy attitudes is misleading as two of the three indirect effects were not significant. The discussion should be based on empirical findings from the study, not speculation of mechanisms unsupported by the data. For example, is there empirical evidence to suggest that SH individuals are more likely to endorse CAM than STEM individuals? If not, the authors need to temper their claims in the discussion.

Another limitation is the correlational, cross-sectional nature of the study. You cannot truly test mediation. Be sure to avoid causal language. Alternative explanations for the findings should also be considered. There are a number of unmeasured factors that may differ between SH and STEM individuals.

Minor Comments

I suggest reformatting the paper to include section headers (e.g., Method).

For all measures, indicate what direction of scores to help readers interpret the data. For example, do higher scores on the vaccine conspiracy attitudes scale indicate more or less vaccine hesitancy?

6. PLOS authors have the option to publish the peer review history of their article (what does this mean?). If published, this will include your full peer review and any attached files.

Reviewer #1: No

Reviewer #2: No

Reviewer #3: No

---

## [Author Response · Author response to Decision Letter 0]

14 Dec 2021

Response to Editor/Reviewers

Dear Editor,

Dear Reviewers,

Thank you for giving us the opportunity to revise and resubmit this manuscript. We appreciate your suggestions and we have incorporated them into the revised manuscript. We hope that paper is significantly improved in this way. All our specific responses are highlighted in blue. 

Reviewer #1: Thank you for the opportunity to review the manuscript entitled ‘Vaccination conspiracy beliefs: differences between persons educated in different scientific fields and their mediating mechanisms'.

The author(s) present a study in which they investigated vaccine conspiracy beliefs in regard to educational background – social science and humanities versus stem. It is an interesting take on the role of education in vaccine conspiracy beliefs, because education is in this kind of research mostly investigated in terms of educational levels and not field differences.

I found the manuscript to be well written, and it contributes with new insights. Nevertheless, I recommend some changes which can contribute to an even greater quality of it.

Title

I suggest renaming the title to emphasize your main finding and make it more interesting for potential readers, some examples are: ‘Vaccine conspiracy beliefs stronger in social sciences and humanities then in STEM – a correlational study’.

Reply: We changed a title to make it clearer, and to emhasize our findings. Thank you for the suggestion.

Abstract

Start with an introduction sentence.

Reply: Added.

Introduction

Page 4, lines 76-79, add references for describing the SH and STEM fields.

Reply: Done.

Throughout the manuscript replace ‘persons’, perhaps with ‘people’, although I am not sure this is the adequate term too.

Reply: Where appropriate (in case where we are talking about the persons who participated in the survey), we changed „persons“ into „participants“. In other cases, we changed „persons“ into „people“.

Research goals, questions, and methods

Please insert a paragraph that in details describes the procedure of how and when the study was conducted - where was the survey shared, which months and year, how the instruments were translated to Croatian.

Reply: we added the requested details, and also explained that the data were taken from the wider pilot study of vaccine hesitancy in Croatia.

Check the journal guidelines – but I find the Descriptive tables should be better put in the Results section. 

Reply: We understand your point, but we kept it in the Methods (now: Methods and instruments) section since they describe the measures employed in the study, i.e. they are not connected to our research questions as such.

In the Descriptive tables, you can write the SD on items in brackets and italic after the M (e.g., M(Sd) 2.32(1.37))

Replace ‘Reverse items’ with ‘Reversely scored items’ throughout the manuscript.

Reply: Done.

Results

Add the age range, apart from the M and Sd, as in minimum and maximum.

Reply: Done.

In Table 6, the significance of the standardized Beta regression coefficients should be written. Now only the statistical significance of B is written.

Reply: In the revised version we did not conduct hierarchical regression analysis, since mediation analysis is done by PROCESS in a more precise way. Standardized coefficients are not part of the regression table any more.

I would be much more interested to see the differences in the used instruments between the two educational groups then the skewness and kurtosis. So please add another table which will show the M and Sd of the used instruments separately for the SH and STEM subsamples. Even conducting an ANOVA would be appropriate.

Reply: we see your point, and that is why we added the separate descriptive statistics for SH and STEM for each instrument. Additionally, we also added intercorrelational matrix, as well as effect sizes (Cohen's d) for nominal variables. All this, we hope, should provide a clearer insight into the variables relatioinships.

Discussion

Start with a sentence that reminds the reader of the aim of the study.

Reply: Done, we start with the brief summary of the results and their relation to our research questions.

At the end, I also think it is important to state the overall limitations of an online survey, your sample and methodology in general.

Reply: Done. 

Highlights

I suggest rewriting the highlights as more comprehendible or fuller sentences, it is hard for me as a reader to follow it in this form.

Reply: Done

Overall

Language and grammar require some editing (especially in the highlights section), so I suggest carefully reading through the manuscript and double checking or engaging a native speaker. I would also emphasize that the variables used here are not the only ones that are shown to be associated with vaccine conspiracy beliefs (another example from Croatia is DOI: 10.1080/08870446.2019.1673894)

Reply: Done

Reviewer #2: This article is exploring the differences in vaccination conspiracy beliefs between people belonging to different scientific fields – Social Sciences & Humanities (SH) and Science, Technology, Engineering and Mathematics (STEM). The results revealed that those educated in SH were more prone to vaccination conspiracy beliefs, and this was mediated by health beliefs; lower trust in the healthcare system, and lower science literacy.

Although the authors deal with important questions for the society today, the rationale for this specific research question is not explored enough. The theoretical background need to be more explained, followed by the importance of differentiation between SH and STEM persons. Is there any specific recommendation coming from these findings (in terms of science education)?

Reply: We added an explanation of the importance of the differentiation between SH and STEM (86-90).

Some issues remained unanswered: What kind of professions exactly, do the authors have in the groups under the umbrella of STEM and SH? 

Reply: Done. We added the following sentence - As for the delineation of SH and STEM fields, the official classification of Ministry of Science and Education of the Republic of Croatia was used. 

What are the employment statuses of the respondents? Sometimes people work in completely different fields then those from which they graduated. It would be good to discuss the relation between employment-education status of the respondents and relation of those variables to the conspiracy belief.

Reply: we completely agree with this, but unfortunately we did not measure employment status, i.e. we do not have this indicator available in the dataset.

Structure of the manuscript in some parts is not adequate for the academic paper. There is no difference between Introduction and Method. Research goals and questions should be placed in the Introduction part. Method part is missing; it should stand alone not incorporated in other parts. Instruments should be placed in the Method part. 

Reply: We now clearly separated research goals and hypotheses from methods, and moved the methods into a separate section (now: Methods and instruments).

Sample information is not sufficient: we need to know when the survey was conducted, whether there was ethical approval from any institution, the length of the questionnaire. Also I would recommend reporting the sample characteristics in more details.

Reply: All the details are added now ()243-254).

Additionally, descriptive statistics are usually part of results not method part. 

Reply: We agree, but in our case descriptive statistics that is presented is not connected to the research questions, but has a role of presenting the characteristics of the measurement instruments.

Endorsement of scientific knowledge and vaccine conspiracy belief should be mentioned as research question (because they are presented in the tables), and the total scores could be given separately for SH and STEM part

Reply: Done, we presented all descriptive statistics separately for SH and STEM, and also added intercorrelational matrix which sheds more light onto the relationships between the variables.

Tables are not according to APA standards.

I would also like to see empirical confirmation for the assumptions mentioned in the discussion: ...Additionally, a large portion of these differences can be explained by the mediating factors that are part of the 3C's model which, according to our claim, might be stemming from the differences in epistemic cultures, i.e. teaching patterns, cognitive approaches, research traditions, goals, behaviors, logic and methods. How can authors assume that? In the discussion part it would be better if the authors analyze results that they measured in the study.

Reply: we also agree with this, that is why in the revised version we emphasized that only science literacy was a statistically significant mediator. But we also pointed out to the fact that confidence intervals in the bootstrapping analyses of the remaining two mediators include zero effect by only a small margin, and that the coefficients go into the direction that is consistent with the assumed epsitemic differences between the science fields. In other words, we recommended that these mediators are worth researching in studies with other (random) samples.

Finally, what is the relation between Conspiracy beliefs vs. Vaccine intentions? What are the effects of conspiracy belief in the current pandemic? And how can the results of the study be connected to the situation in pandemic – covid-19 conspiracy theories. The connection between vaccine CT and Covid-19 CT should be explained.

Reply: We added the commentary of the aplicabiltiy of our results to the management of COVID-19 pandemic. In brief, we pointed out to the fact that vaccination conspiracy beliefs are strongly connected to health beliefs (importance of natural immunity) and to the trust in healthcare system (this can be seen from the intercorrelation matrix and from the results of regression analysis), i.e. that that vaccine hesitancy cannot be reduced solely by providing factual information.

Reviewer #3: The goal of the present study was to determine whether were differences in vaccine hesitancy and likelihood of accepting vaccine conspiracy theories in individuals educated in social sciences and humanities versus those educated in STEM. The research question is interesting and timely given concerns related to COVID-19 vaccine hesitancy. However, the rationale is underdeveloped, the method/results lack important details, and the discussion over-reaches. Below, I outline my concerns in more detail.

The introduction is underdeveloped. The authors mention the 3Cs at the beginning of the introduction and review some research on the “confidence” component, but no attention is given to convenience or complacency. They then state that they will examine the 3Cs in relation to those educated in STEM and social sciences/humanities. It is unclear how or why the 3Cs are being examined.

Reply: we agree, but 3Cs model was described only to point out that our mediators are not chosen at random, but that they are a part of a wider picture which can be systematized by 3Cs model. We added explanation which elements of 3Cs model we are using - institutional trust (confidence), postmodern health beliefs (complacency), and science literacy (convenience - “ability to understand”).

More discussion and connection to the research questions would be helpful. The authors also outline differences between STEM and social sciences/humanities fields, but don’t clearly connect this to vaccine hesitancy. It would be helpful for the authors to more explicitly state their reasoning and provide hypotheses, if they had a priori hypotheses. Clearer rationale for the mediation analyses needs to be provided.

Reply: we hope that we adressed these concerns in the revised version of the manuscript. We did transform or research questions into hypotheses, because we indeed posed hypotheses based on our reflection of epistemic differences between SH and STEM.

More information needs to be provided regarding participant recruitment. How were participants contacted? What was the response rate? Is the sample representative of the population in question? What range of degrees are represented in the sample (e.g., BS/BA, MS/MA, PhD, etc.)? Did participants receive compensation? Was there informed consent?

Were there any missing data, outliers, non-normality, etc.? If so, how were these issues addressed?

Reply: We added more information about sample and recruitment (243-254). There were not significant outliers, while non-normality should be successfully handled by PROCESS according to Hayes book (Introduction to Mediation, Moderation, and Conditional Process Analysis, 2018, New York: Guilford Press). As for missing values, about 7,53% of all cases did have missing values on at least one of the study variables. We conduced binary logistic regression with the missing value as the criterion variable and did not find any significant relationship with other study variables.That is why we proceeded with the complete case analysis, which reduced our sample from 624 do 577 respondents. The description is now added into the revised version of the manuscript.

Why were age, gender, and religiosity included as covariates? Did results differ if covariates were excluded?

Reply: These covariates were chosen because of the possible connection to vaccine hesitancy based on the previous studies. The results did differ with regards to gender, and this was an excellent point that we now added to the discussion of the results.

Before reporting the mediation analysis, it would be useful to report simple comparisons (t-tests) indicating whether STEM vs. SH participants differed in any of the mediators or outcome variables.

Reply: This is a very good point, and it was also suggested by other reviewers. We did this by adding the intercorrelation matrix and effect sizes (Cohen's d) of the nominal variables. Hopefully, this shed more light onto the relations between the study variables before the mediation analysis.

A clearer description of the mediation analysis(es) is necessary, noting how variables were coded (i.e., STEM vs. SH) and included in the model. I assume a parallel mediation model was estimated.

Reply: Yes, it as a multiple parallel mediation. We added a conceptual diagram that should make the entire procedure much clearer than in the initial version of the manuscript.

Remove reference to marginal significance (p = .06). If relying on p-values, an effect is significant or not. Also, discussion of full or partial mediation is inappropriate, given the correlational and cross-sectional nature of the study.

Reply: Yes, we avoided this reference. But having in mind that confidence intervals include zero effect by only a small margin, form the point of view of Bayesian statistics as well for theoretical reasons (all mediation coefficients point to the hypothesized direction), we just emphasized that the suggested mediating mechanisms are worth of researching in future studies with random samples.

It is unclear to me what purpose the hierarchical regression analysis serves.

Reply: PROCESS is a regression-based method and hierarchical regression is also used to study mediation. However, PROCESS is a strict method that gives confidence intervals for direct and indirect effects, while hierarchical regression is much more „impressionistic“ method, i.e. it does not include explict test of the effects. That why we followed your suggestion and removed the hierarchical regression. As noted PROCESS is regression-based, so we provided the overall regression that is a part of the PROCESS output. This is useful because it shows the direct effects of all predictors. The other part of the story (overall and indirect effects of the field of study) is presented in the mediation analysis. Hopefully, we made it all clearer in the revised manuscript.

The claim in the discussion that the 3Cs mediated difference in education field and vaccine conspiracy attitudes is misleading as two of the three indirect effects were not significant. The discussion should be based on empirical findings from the study, not speculation of mechanisms unsupported by the data. For example, is there empirical evidence to suggest that SH individuals are more likely to endorse CAM than STEM individuals? If not, the authors need to temper their claims in the discussion.

Reply: We agree, and we made it excplicit now what the study results showed (only one of the mediation paths is statistically significant), and what are our hypotheses.

Another limitation is the correlational, cross-sectional nature of the study. You cannot truly test mediation. Be sure to avoid causal language. Alternative explanations for the findings should also be considered. There are a number of unmeasured factors that may differ between SH and STEM individuals.

Reply: we fully agree, that is why we included demographic control variables. And we are also aware that the study is correlational, we added it as an additional limitation at the end of the paper.

Minor Comments

I suggest reformatting the paper to include section headers (e.g., Method).

For all measures, indicate what direction of scores to help readers interpret the data. For example, do higher scores on the vaccine conspiracy attitudes scale indicate more or less vaccine hesitancy?

Reply: Added.

---

## [Decision Letter · Decision Letter 1]

14 Jan 2022

PONE-D-21-19906R1Vaccination conspiracy beliefs among social science & humanities and STEM educated people - an analysis of the mediation pathsPLOS ONE

Dear Dr. Šuljok,

Thank you for submitting your manuscript to PLOS ONE. After careful consideration, we feel that it has merit but does not fully meet PLOS ONE’s publication criteria as it currently stands. Therefore, we invite you to submit a revised version of the manuscript that addresses the points raised during the review process.

We look forward to receiving your revised manuscript.

Kind regards,

Steven Frisson

Academic Editor

PLOS ONE

Journal Requirements:

Additional Editor Comments:

Editor’s letter PONE-D-21-19906-R1

Please note that I was not the Editor for the original submission. However, the same Reviewers who assessed the original submission also reviewed the revision. I read the manuscript carefully myself (I have done some related research lately) and think it is an interesting study that will be of interest to the readers of PLOS One.

My decision based on the three reviews is that a “minor revision” is required for the present manuscript. While Reviewers 1 and 2 recommend acceptance, Reviewer 3 makes a number of important points, some of which mirroring my own take on the manuscript (e.g. point 4).

Please address all the Reviewer’s comments in your revision, as well as the following (mainly error corrections) – numbers correspond to line numbering:

- throughout: change “persons” to “people”

- throughout: be consistent in your use of tense. Often past and present tenses are mixed, even in the same paragraph (e.g. paragraph starting line 176, paragraph starting line 281), which is very confusing for the reader.

- 45: “become a commonplace” is hardly used in English, change to “has become commonplace”

- 56: “Various research” change to “Research” and add some references.

- 62 and 486: “and healthcare system” change to “and the healthcare system”

- 76: “vaccine hesitance” change to “vaccine hesitancy”

- 81: add comma after “[28]”

- 101: I find the term “socialization” somewhat problematic as it seems to suggest that it is an exclusively external influence (basically, all “nurture”) without allowing the fact that certain personality traits make people more or less likely to pursue a “soft” or “hard” science. Ideally, one would want to carry out a study amongst final year secondary school students who are planning to go into either direction.

- 112: “there only few” change to “there are only few”

- 113: “differences” – unclear what the directions of the differences are.

- 125-129: please rewrite the sentence as I am unsure what is being stated.

- 139: “on the sample” change to “on a sample”

- 151-155: another difference between SH and STEM is a difference in focus, with STEM more focused on groups and averages and SH on individuals.

- 165: willing to accepted” change to “willing to accept”

- 204: you have to start with the participant info otherwise the results in Tables 1 - 4 are uninterpretable.

- as suggested by one of the Reviewers in the previous round, please add simple t-tests when splitting the data between groups, e.g. lines 209-210).

- 210, 220, 233: “higher results” change to “higher values”

- Table headings for Tables 1 to 4: “for THE… scale”

- 237: please add a reference for the Oxford scale

- 249: please add references for the surveys referred to.

- 266: “consisted of the college”: delete “the”

- 269: “conduced” change to “conducted”

- 277: add a reference or link for the classification used

- 303: delete “Andrew F.”

- Tables 6 & 7: define LLCI and ULCI

- paragraph starting 308: why are the numbers (e.g. 1.36) different from those in the Table?

- 315: “is statistically” change to “is a statistically”

- 337: “significant”: where is this indicated in the Table?

- 352: “with the other” change to “with other”

- 372: “additional analyses”: where can the reader find these?

- 382: delete “so”

- 385-386: “sample of…”: please rephrase

- 428: “found…correlation”: change to either “found a ... correlation” or “found… correlations”

- 430: “indicate” change to “indicates”

- 432: “having in mind that” change to “in that”

- 432: “the vaccine” change to “vaccine”

- 455: “to much” change to “too much”

- 456: “object the fact” change to “object to the fact”

- 471: “as well with” change to “as well as with”

- 494: delete “in it”

Just as a suggestion, I think some of the research by Gordon Pennycook and colleagues will be of interest to your future research, e.g. their examination of a bias to accept “pseudo-profound bullshit”, which is linked with e.g. religious beliefs and analytical thinking (e.g. Pennycook & Rand, 2019; DOI: 10.1111/jopy.12476).

Reviewers' comments:

Reviewer's Responses to Questions

**Comments to the Author**

1. If the authors have adequately addressed your comments raised in a previous round of review and you feel that this manuscript is now acceptable for publication, you may indicate that here to bypass the “Comments to the Author” section, enter your conflict of interest statement in the “Confidential to Editor” section, and submit your "Accept" recommendation.

Reviewer #1: All comments have been addressed

Reviewer #2: All comments have been addressed

Reviewer #3: (No Response)

2. Is the manuscript technically sound, and do the data support the conclusions?

Reviewer #1: Yes

Reviewer #2: Yes

Reviewer #3: Partly

3. Has the statistical analysis been performed appropriately and rigorously? 

Reviewer #1: I Don't Know

Reviewer #2: Yes

Reviewer #3: Yes

4. Have the authors made all data underlying the findings in their manuscript fully available?

Reviewer #1: Yes

Reviewer #2: Yes

Reviewer #3: Yes

5. Is the manuscript presented in an intelligible fashion and written in standard English?

Reviewer #1: Yes

Reviewer #2: Yes

Reviewer #3: Yes

6. Review Comments to the Author

Reviewer #1: Thank you for the opportunity to review the revised manuscript entitled Vaccination conspiracy beliefs among social science & humanities and STEM educated people - an analysis of the mediation paths, and for addressing all my suggestions.

Reviewer #2: (No Response)

Reviewer #3: Overall, the authors have done a nice job addressing my initial concerns and suggestions. The manuscript is much stronger. I do have a few minor suggestions.

1. In the introduction, it would be helpful to explicitly connect field of study with the 3Cs to clearly articulate rationale.

2. There are still a few places in the manuscript where the authors refer to "causal" relationships. That cannot be determined from this study.

3. Provide an explicit rationale for the inclusion of covariates in the analyses. Also, as the some of the results were different when covariates were excluded, report the stats in text or supporting material.

4. It is important to acknowledge that differences between education and training in these fields may not be the reason for these findings. Rather, there may be unmeasured individual differences (e.g., personality, cognitive styles, etc.) that lead people into different educational fields and influence vaccination conspiracy beliefs.

7. PLOS authors have the option to publish the peer review history of their article (what does this mean?). If published, this will include your full peer review and any attached files.

Reviewer #1: No

Reviewer #2: No

Reviewer #3: No

---

## [Author Response · Author response to Decision Letter 1]

26 Jan 2022

Dear Editor, 

Thank you for giving us the opportunity to revise and resubmit new version of this manuscript. We appreciate the time and suggestions provided by reviewer 3 and by you and we have incorporated the suggested changes into the manuscript.

We have responded specifically to each suggestion below, beginning with your own. 

- throughout: change “persons” to “people”

Response: Done

- throughout: be consistent in your use of tense. Often past and present tenses are mixed, even in the same paragraph (e.g. paragraph starting line 176, paragraph starting line 281), which is very confusing for the reader.

Response: Past tense is now consistently used throughout the paper.

- 45: “become a commonplace” is hardly used in English, change to “has become commonplace”

- 56: “Various research” change to “Research” and add some references.

Response: Done.

- 62 and 486: “and healthcare system” change to “and the healthcare system”

Response: Done.

- 76: “vaccine hesitance” change to “vaccine hesitancy”

Response: Done.

- 81: add comma after “[28]”

Response: Done.

- 101: I find the term “socialization” somewhat problematic as it seems to suggest that it is an exclusively external influence (basically, all “nurture”) without allowing the fact that certain personality traits make people more or less likely to pursue a “soft” or “hard” science. Ideally, one would want to carry out a study amongst final year secondary school students who are planning to go into either direction.

Response: We agree. The term „socialization“ is changed to „background“, since the latter term covers both the academic socialization and the preexisting differences among the students. Also we added sentence „Such differences can be the result of the different academic socializations and/or the individual characteristics which are important when choosing field of study.” Also in chapter Conclusion and limitations „The reasons could probably be found in the different academic socialization and the background characteristics which are important when selecting a field of study.“

- 112: “there only few” change to “there are only few”

Response: Done.

- 113: “differences” – unclear what the directions of the differences are.

Response: The direction is now explicitly stated („mainly confirming that vaccine hesitancy is higher among SH graduates“). 

- 125-129: please rewrite the sentence as I am unsure what is being stated.

Response: Done.

- 139: “on the sample” change to “on a sample”

Response: Done.

- 151-155: another difference between SH and STEM is a difference in focus, with STEM more focused on groups and averages and SH on individuals.

Response: We agree, but that topic was not covered in the research that we refer to. 

- 165: willing to accepted” change to “willing to accept”

Response: Done.

- 204: you have to start with the participant info otherwise the results in Tables 1 - 4 are uninterpretable.

Response: The participants' info is moved to the beginning of the chapter.

- as suggested by one of the Reviewers in the previous round, please add simple t-tests when splitting the data between groups, e.g. lines 209-210).

Response: Done.

- 210, 220, 233: “higher results” change to “higher values”

Response: Done.

- Table headings for Tables 1 to 4: “for THE… scale”

Response: Done.

- 237: please add a reference for the Oxford scale

Response: Done.

- 249: please add references for the surveys referred to.

Response: Done.

- 266: “consisted of the college”: delete “the”

Response: Done.

- 269: “conduced” change to “conducted”

Response: Done.

- 277: add a reference or link for the classification used

Response: Done.

- 303: delete “Andrew F.”

Response: Done.

- Tables 6 & 7: define LLCI and ULCI

Response: Explanations added.

- paragraph starting 308: why are the numbers (e.g. 1.36) different from those in the Table?

Response: It is corrected now.

- 315: “is statistically” change to “is a statistically”

Response: Done.

- 337: “significant”: where is this indicated in the Table?

Response: It is visible since the confidence interval includes zero (effect).

- 352: “with the other” change to “with other”

Response: Done.

- 372: “additional analyses”: where can the reader find these?

Response: We included them in the paper now (i.e. the magnitude of the indirect effects when gender is not a covariate). All analyses can be easily replicated by using our data and SPSS PROCESS syntax (process y=Vac_con /x=Stu_re /m=Lit Imm HCS /cov= Age Relig Gender /model=4.) Since PROCESS uses bootstraping results can be negligibly different from analysis to analysis, but the conclusions are always the same (we repeated all analyses several times).

- 382: delete “so”

Response: Done.

- 385-386: “sample of…”: please rephrase

Response: It is rephrased as: „as well as with a research conducted on a sample of Croatian population”.

- 428: “found…correlation”: change to either “found a ... correlation” or “found… correlations”

Response: Done.

- 430: “indicate” change to “indicates”

Response: Done.

- 432: “having in mind that” change to “in that”

Response: Done.

- 432: “the vaccine” change to “vaccine”

Response: Done.

- 455: “to much” change to “too much”

Response: Done.

- 456: “object the fact” change to “object to the fact”

Response: Done.

- 471: “as well with” change to “as well as with”

Response: Done.

- 494: delete “in it”

Response: Done.

Just as a suggestion, I think some of the research by Gordon Pennycook and colleagues will be of interest to your future research, e.g. their examination of a bias to accept “pseudo-profound bullshit”, which is linked with e.g. religious beliefs and analytical thinking (e.g. Pennycook & Rand, 2019; DOI: 10.1111/jopy.12476).

Response: Than your for the suggestion, we are going to use it in the continuation of our project.

Reviewer #3: 

Reviewer #3: Overall, the authors have done a nice job addressing my initial concerns and suggestions. The manuscript is much stronger. I do have a few minor suggestions.

1. In the introduction, it would be helpful to explicitly connect field of study with the 3Cs to clearly articulate rationale.

Response: the following sentence is added: „. Additionally, it is our contention that such differences will be mediated by some of the indicators coming from the 3Cs model (trust in the healthcare system, specific health beliefs, and science literacy).”

2. There are still a few places in the manuscript where the authors refer to "causal" relationships. That cannot be determined from this study.

Response: We checked it, we use this word only in the following sentence: „In addition, given that our study was cross-sectional and the possible omitted variables bias might have existed, causal mechanisms cannot be fully supported by the data from our study.“. Following suggestions of the reviewer, in other parts of the paper we only refer to „correlations“, „connections“, etc. Please let us know if we are wrong, i.e. if we missed some of the causal implying language. 

3. Provide an explicit rationale for the inclusion of covariates in the analyses. Also, as the some of the results were different when covariates were excluded, report the stats in text or supporting material.

Response: The following sentence is added: „. The covariates were used in order to check for the possible spurious effects of our predictors.”.

The stats (indirect effects) of the model when gender is not added as a covariate are now reported in the text.

4. It is important to acknowledge that differences between education and training in these fields may not be the reason for these findings. Rather, there may be unmeasured individual differences (e.g., personality, cognitive styles, etc.) that lead people into different educational fields and influence vaccination conspiracy beliefs.

Response: Yes, we agree. That is why we replaced the term „academic socialization“ with „academic background“, since the latter term implies both possible explanations. The following sentence is also added: „Such differences can be the result of the different academic socializations and/or the individual characteristics which are important when choosing a field of study.”(in Introduction) and „The reasons could probably be found in the different academic socialization and the background characteristics which are important when selecting a field of study.“ (in chapter Conclusion and limitations).

---

## [Editor Report · Decision Letter 2]

16 Feb 2022

Vaccination conspiracy beliefs among social science & humanities and STEM educated people - an analysis of the mediation paths

PONE-D-21-19906R2

Dear Dr. Šuljok,

We’re pleased to inform you that your manuscript has been judged scientifically suitable for publication and will be formally accepted for publication once it meets all outstanding technical requirements.

Kind regards,

Steven Frisson

Academic Editor

PLOS ONE

Additional Editor Comments:

Please make the following minor corrections:

- line 50: comma after "MacDonald[30]"

- line 131: comma after [5] and before "Biglan's"

- line 208: I don't understand the use of "whereas" here as it is not contrastive. Change to "and"

- line 218: change "classification of Ministry" to "classification by the Ministry"

- lines 223-224: change "agreed that project" to "agreed that the project"

- line 404: change "that bootstrapping" to "that the bootstrapping"

- line 406: change "in healthcare system" to "in the healthcare system"

- line 413: change "aware of distinctiveness" to "aware of a distinctiveness"

Reviewers' comments:

N/A

---

## [Editor Report · Acceptance letter]

21 Feb 2022

PONE-D-21-19906R2 

Vaccination conspiracy beliefs among social science & humanities and STEM educated people - an analysis of the mediation paths 

Dear Dr. Šuljok:

I'm pleased to inform you that your manuscript has been deemed suitable for publication in PLOS ONE. Congratulations! Your manuscript is now with our production department. 

Kind regards, 

on behalf of

Dr. Steven Frisson 

Academic Editor

PLOS ONE